# *GeoMoLa*: Geometry-Aware Motion Latents for Learning Robust Manipulation Policies

## Abstract

Learning motion latents for robotic manipulation heavily relies on extracting motion patterns from visual sequences, yet effective action abstractions require understanding three-dimensional geometric transformations. Here, we introduce *GeoMoLa* (Geometry-Aware Motion Latents), which learns discrete motion latent codes by predicting how point clouds evolve during manipulation rather than reconstructing visual observations. This four-dimensional objective – spatial geometry changing through time – forces latent representations to encode actual physical motion rather than appearance patterns. *GeoMoLa* achieves state-of-the-art performance using only single-view RGB-D input, while existing methods require multi-view reconstruction, succeeding across diverse manipulation benchmarks. Our ablations reveal that geometric prediction is the key to driving performance, quantitatively validating that manipulation depends on spatial understanding. Furthermore, the learned codes exhibit effective motion abstraction: applying them to novel scenes produces physically consistent transformations regardless of visual context. Our real-world experiments also confirm this robustness capability, achieving robust manipulation with minimal demonstrations in cluttered environments where geometric reasoning determines success. Thus, we demonstrate that effective motion latents for robot control can better emerge from understanding motion through its three-dimensional effects rather than pixel-level patterns.

## 1 Introduction

Robot manipulation requires learning reusable motion patterns – motion latents (Bruce et al., 2024; Parker-Holder et al., 2024; Ball et al., 2025) – that abstract complex continuous movements into discrete, transferable skills. Current methods mainly learn these motion latents from sequences of two-dimensional images, missing the three-dimensional geometric structure that fundamentally determines manipulation success. A grasping action, for instance, depends not only on visual appearance but also on precise spatial relationships, approach angles, and the continuous evolution of three-dimensional configurations over time.

*Therefore,* learning motion latents without access to this underlying spatiotemporal geometry may produce representations that fail to generalize across different viewpoints, object poses, or spatial arrangements. *Furthermore,* this representational gap could create cascading failures in real-world deployment. Robots may not recognize that occluded objects maintain their geometric relationships despite visual changes, and small spatial errors compound across action sequences without understanding of three-dimensional workspace dynamics. Solving this essential representation issue in motion latent learning is necessary for robots to understand manipulation through spatial relationships and physical transformations rather than pixel patterns.

The *core* challenge lies in jointly modeling spatial geometry and temporal dynamics without prohibitive computational cost. Existing methods capture either spatial structure through static three-dimensional representations (Ke et al., 2024; Ze et al., 2024) or temporal patterns through two-dimensional video (Ye et al., 2024; Chen et al., 2024), but not both. Three-dimensional approaches process frozen point clouds without modeling evolution; diffusion policies generate trajectories from fixed scene features; video-based latent learning operates in image space without depth.

Our *key* insight is that effective motion latents must encode geometric transformations in three-dimensional space over time, not static scenes or visual motion. In this paper, we propose *GeoMoLa*

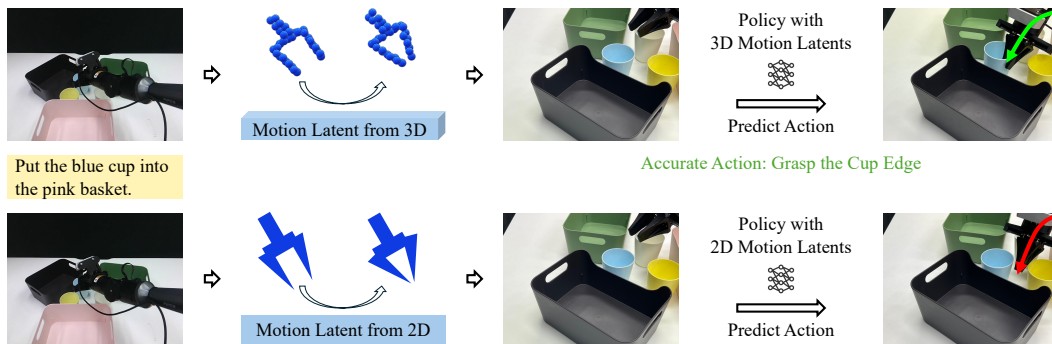

Figure 1: **Policies trained with 2D and 3D motion latents.** When encountering novel and cluttered scenes (left) that differ from the training distribution, the policy trained with 3D motion latents (top) demonstrates superior task performance with more robust control, e.g., enhanced reaching and grasping accuracy.

(Geometry-Aware Motion Latents), which learns 3D discrete motion latent codes (i.e., motion latent) by predicting how 3D point clouds or point maps evolve during manipulation. By training latent representations to forecast future geometric states rather than reconstruct current observations, we ensure these codes capture the causal relationship between actions and their spatial effects, creating motion primitives grounded in physical transformation rather than visual appearance.

Our approach demonstrates that explicitly modeling four-dimensional dynamics through geometry-aware motion latents significantly improves manipulation performance. We achieve state-of-the-art results using only single-view RGB-D input, while our learned latent codes produce consistent geometric transformations across different scenes, validating their transferability. Ablation studies reveal that three-dimensional geometric prediction is critical while visual appearance contributes minimally, suggesting rethinking representation learning for manipulation. In real-world experiments with limited demonstrations, our method excels particularly in cluttered environments where spatial reasoning determines success. While focused on rigid-body manipulation, our framework establishes four-dimensional geometric prediction as an effective objective for motion latent learning for robotics.

In summary, we make the following contributions:

- We introduce the first framework that explicitly models robot manipulation as continuous four-dimensional processes while learning geometry-aware motion latents through self-supervised prediction of future three-dimensional point cloud evolution.
- We demonstrate through comprehensive ablations that geometric structure in the motion latents is significantly more important than visual appearance for manipulation success.
- We achieve state-of-the-art performance across multiple benchmarks while showing that learned motion latents transfer consistently across different scenes and configurations.
- We show superior real-world performance with limited demonstrations, particularly excelling in cluttered scenarios requiring precise spatial reasoning.

## 2 RELATED WORK

**Motion latents.** Learning motion latent representations has proven effective across diverse applications. GENIE (Bruce et al., 2024; Parker-Holder et al., 2024; Ball et al., 2025) maps user inputs to latent spaces for generating interactive environments, while ILPO (Edwards et al., 2019) employs motion latents for pretraining video game policies. Recent work has explored deriving motion latents directly from observations: LAPA (Ye et al., 2024) and Moto (Chen et al., 2024) extract motion latents from raw inputs to leverage unlabeled data at scale. However, these observation-based approaches overlook the inherent 4D spatiotemporal structure of robotic actions. In contrast, we derive geometry-aware motion latents from 4D trajectories that explicitly capture spatial deformations and temporal dynamics, enabling more robust transfer to robotic manipulation tasks.

**Diffusion models in robotics.** Diffusion models have emerged as powerful tools for robotic manipulation, particularly for trajectory generation and action prediction. ChainedDiffuser (Xian et al., 2023) replaces traditional motion planners with a trajectory diffusion model that conditions on 3D scene

features and predicted keyposes from Act3D (Gervet et al., 2023) to generate linking trajectories. Building on this, 3D Diffuser Actor (Ke et al., 2024) tackles the more challenging task of jointly predicting the next keyposes and linking trajectories, while 3D Diffusion Policy (Ze et al., 2024) combines 3D representations with diffusion objectives. We also evaluate against recent diffusion-based methods (Bu et al., 2024; Black et al., 2023) and demonstrate superior performance. Unlike these approaches that primarily focus on static 3D representations, our method explicitly models 4D dynamics through motion latents, enabling better temporal reasoning and generalization.

**2D and 3D scene representations for robot manipulation.** 3D scene-to-action policies address this limitation through explicit geometric reasoning: C2F-ARM (James et al., 2021) and PerAct (Shridhar et al., 2022) voxelize workspaces but face computational scaling challenges; Act3D (Gervet et al., 2023) avoids voxelization by sampling and featurizing 3D points through cross-attention; and RVT (Goyal et al., 2023) reprojects RGB-D inputs to multiple views before lifting predictions to 3D. End-to-end image-to-action models like RT-1 (Brohan et al., 2022), RT-2 (Brohan et al., 2023), GATO (Reed et al., 2022), BC-Z (Jang et al., 2022), RT-X (Padalkar et al., 2023), Octo (Team et al., 2024), and InstructRL (Liu et al., 2022) directly predict 6-DoF poses from 2D images but require thousands of demonstrations to implicitly learn 3D geometry. While these methods improve upon 2D approaches through explicit 3D representations, they still treat manipulation as static spatial reasoning. Our approach advances beyond static 3D by modeling actions as continuous 4D processes, capturing how spatial configurations evolve over time.

# 3 METHOD

**Overview.** We aim to learn robotic manipulation policies that map RGB-D observations and task instructions to executable actions. Our *key* contribution is learning discrete motion latents – *abstract motion concepts that guide high-level policy planning* – from 4D spatiotemporal data (3D pointmaps over time) rather than 2D video sequences. Motion latents are essential for abstracting reusable motion patterns and enabling task generalization, yet most existing methods learn motion latents directly from 2D observations, missing crucial 3D geometric information (depth relationships, spatial arrangements, and object poses) that fundamentally determine manipulation feasibility.

Our geometry-aware motion latents address this limitation by encoding abstract motion concepts grounded in 3D geometry. This approach provides three advantages: better generalization across manipulation tasks through abstract motion primitives, improved interpretability by encoding 3D geometric transformations, and enhanced performance in cluttered environments requiring precise spatial reasoning. Our framework comprises two components: a self-supervised pipeline that discovers geometry-aware motion latents by predicting future 3D observations from demonstrations, and a diffusion-based model that leverages these 3D action latents to generate actions.

## 3.1 PROBLEM FORMULATION

We consider a dataset of robotic manipulation demonstrations $\mathcal{D} = \{(\{\mathbf{o}_i^t, \mathbf{a}_i^t\}_{t=1}^{T_i}, l_i)\}_{i=1}^N$, where each demonstration contains a sequence of observation-action pairs with a natural language instruction $l_i$. Each observation $\mathbf{o}_i^t = (\mathbf{I}_i^t, \mathbf{D}_i^t)$ consists of an RGB image $\mathbf{I}_i^t \in \mathbb{R}^{H \times W \times 3}$ and a depth map $\mathbf{D}_i^t \in \mathbb{R}^{H \times W}$. Each action $\mathbf{a}_i^t = (\mathbf{p}_i^t, \mathbf{r}_i^t, g_i^t)$ specifies the end-effector position $\mathbf{p}_i^t \in \mathbb{R}^3$, rotation $\mathbf{r}_i^t \in \mathbb{R}^6$, and gripper state $g_i^t \in \{0, 1\}$. We employ the 6D rotation representation proposed by Zhou et al. (2018) to circumvent the discontinuity issues inherent in quaternion representations. Our objective is to learn a policy $\pi$ that maps the observation history and task instruction to an action chunk: $\hat{\mathbf{a}}^{t:t+h-1} = \pi(\mathbf{o}^t, l)$, where $h$ is the prediction horizon. The challenge lies in capturing both geometric manipulation constraints (e.g., collision avoidance, grasp stability) and semantic language intent. We address this through a two-stage method: First, we learn a dictionary of geometry-aware motion latents (i.e., motion latents) that capture reusable motion patterns from demonstrations (Sec. 3.2). Second, we leverage these learned motion latents to guide a diffusion-based policy for generating executable trajectories (Sec.3.3). We now detail each component.

## 3.2 LEARNING MOTION LATENTS

We propose a self-supervised framework that discovers motion patterns by learning a dictionary of discrete 3D latent codes, referred to as motion latents, encoded from the current observation and task instructions, and trained to predict future 3D observations. We predict the future 3D geometry to encourage the motion latents to capture the underlying motions driving scene evolution. We use discrete rather than continuous representations because they naturally cluster similar motions into reusable primitives and provide interpretable, composable action abstractions.

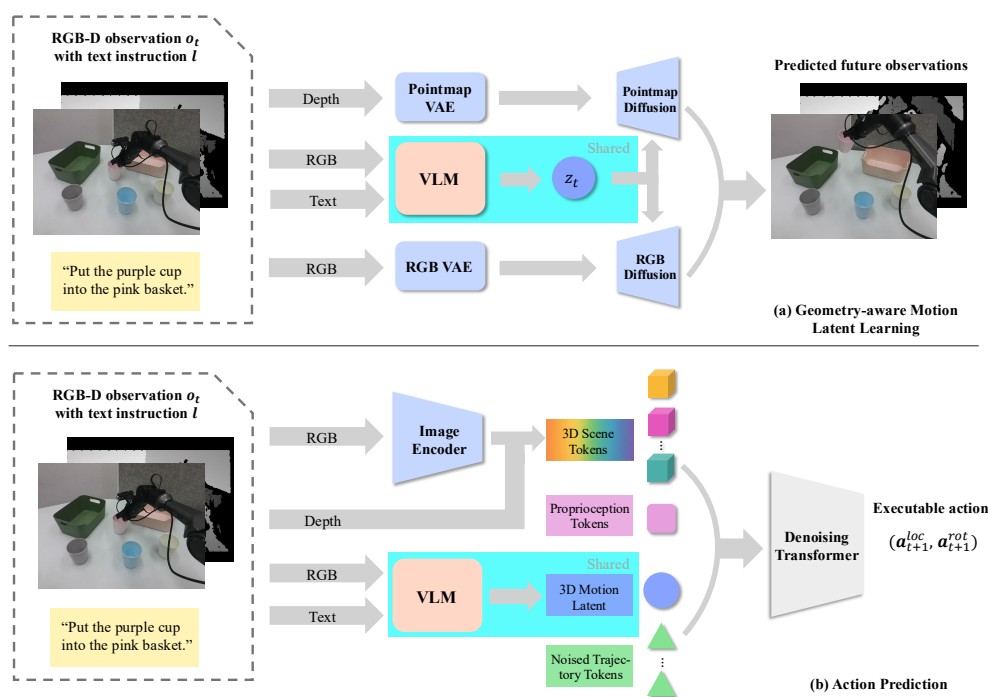

Figure 2: *GeoMoLa* **framework. (a)** Geometry-Aware Motion Latent Learning: RGB-D observations and language instructions are encoded into discrete motion latents via VQ-VAE, trained by predicting future pointmaps and RGB images. This self-supervised objective ensures latent codes capture 4D dynamics (3D geometry over time). **(b)** **Latent-Conditioned Action Prediction:** The previously trained motion latent encoder and the codebook are applied to guide a 3D denoising transformer to generate 6-DoF trajectories through iterative refinement, using 3D-aware attention mechanisms to accommodate geometric constraints.

### 3.2.1 ENCODING OBSERVATIONS INTO 3D MOTION LATENT

Given a current observation $\mathbf{o}^t = (\mathbf{I}^t, \mathbf{D}^t)$ and task instruction $l$, we first extract a continuous representation using a vision-language encoder $\phi^{\text{vlm}}$ based on Mini-GPT (Zhu et al., 2023), which provides strong visual-linguistic grounding: $\mathbf{f}^t = \phi^{\text{vlm}}(\mathbf{o}^t, l)$. We then discretize this representation using Vector Quantization (VQ-VAE) (van den Oord et al., 2017), where we map $\mathbf{f}^t$ to a sequence of $n_s$ discrete codes from a learned codebook $\mathcal{C} = \{\mathbf{c}_1, \mathbf{c}_2, \ldots, \mathbf{c}_K\}$ with vocabulary size $K$:

$$\mathbf{z}^t = \text{VQ}(\mathbf{f}^t) = [z_1^t, z_2^t, \ldots, z_{n_s}^t], \tag{1}$$

where each $z_j^t \in \{1, 2, \ldots, K\}$ is an index into the codebook. We employ NSVQ (Vali & Bäckström, 2022) for stable training, addressing gradient collapse issues common in VQ-VAE.

### 3.2.2 LEARNING VIA FUTURE 3D PREDICTION

The core challenge in learning motion latents is ensuring they encode motion semantics rather than visual appearance. By requiring latent codes to predict future geometric transformations, we force them to capture the causal relationship between actions and their effects in 3D space. Specifically, we train the latent codes $\mathbf{z}^t$ through a conditional prediction task: given the current 3D scene and a latent code, predict how the scene evolves over future timesteps. We use 3D pointmaps to ensure the codes learn geometric transformations rather than pixel-level appearance changes.

**Pointmap Representation.** We convert each RGB-D observation $\mathbf{o}^t$ to a pointmap $\mathbf{P}^t \in \mathbb{R}^{H \times W \times 3}$ by back-projecting pixels to 3D coordinates using camera intrinsics and extrinsics.

**Conditional Diffusion for Future Prediction.** We then employ a video latent diffusion architecture (Blattmann et al., 2023) adapted for pointmaps. The model consists of two components:

(1) A 3D-aware VAE with encoder $\psi^{\text{enc}}$ and decoder $\psi^{\text{dec}}$ that translates pointmaps to and from a latent space: $\mathbf{h}_{\text{pm}}^{t'} = \psi^{\text{enc}}(\mathbf{P}^{t'})$, $\hat{\mathbf{P}}^{t'} = \psi^{\text{dec}}(\hat{\mathbf{h}}_{\text{pm}}^{t'})$, where $\mathbf{h}_{\text{pm}}^{t'} \in \mathbb{R}^d$ denotes the latent encoding at timestep $t'$. The VAE is initialized from a pre-trained RGB VAE (Blattmann et al., 2023) and fine-tuned on pointmaps through reconstruction. See Appendix D for details.

(2) A diffusion model $\psi^{\text{diff}}$ that generates future latents conditioned on history and the motion latent $\mathbf{z}^t$, following DDPM (Ho et al., 2020):

$$\hat{\mathbf{h}}_{\text{pm}}^{t+1:t+w} = \psi^{\text{diff}}(\mathbf{h}_{\text{pm}}^{t-w+1:t}, \mathbf{z}^t; \phi), \tag{2}$$

where $w$ is the observation window size. During training, we minimize the denoising objective: $\mathcal{L}_{\text{diff}}^{\text{pm}} = \mathbb{E}_{k,\epsilon}\left[\|\epsilon - \epsilon_\phi(\mathbf{h}_k, k, \mathbf{h}_{\text{pm}}^{t-w+1:t}, \mathbf{z}^t)\|^2\right]$, where $\mathbf{h}_k = \sqrt{\alpha_k} \cdot \mathbf{h}_{\text{pm}}^{t+1:t+w} + \sqrt{1 - \alpha_k} \cdot \epsilon$ is the noised latent at diffusion step $k$, $\epsilon \sim \mathcal{N}(0, I)$, and $\alpha_k$ follows the noise schedule.

We jointly train an RGB prediction branch using the same architecture and motion latent $\mathbf{z}^t$ to capture correlated appearance changes, with denoising objective $\mathcal{L}_{\text{diff}}^{\text{rgb}}$. The combined training objective is: $\mathcal{L}_{\text{total}} = \mathcal{L}_{\text{diff}}^{\text{pm}} + \mathcal{L}_{\text{diff}}^{\text{rgb}} + \mathcal{L}_{\text{vq}}$, where $\mathcal{L}_{\text{vq}} = \|\text{sg}[\mathbf{f}^t] - \mathbf{c}\|_2^2 + \beta\|\mathbf{f}^t - \text{sg}[\mathbf{c}]\|_2^2$ is the VQ-VAE loss with stop-gradient operator $\text{sg}[\cdot]$ and commitment coefficient $\beta$. Through this objective, latent codes encoding similar motions converge to similar discrete values, creating a learned vocabulary of reusable, geometry-aware motion primitives that capture the essential dynamics of state transitions.

## 3.3 ACTION PREDICTION WITH MOTION LATENTS

Having learned motion latents through self-supervised future prediction, We now describe how to use learned geometry-aware motion latents to generate executable robot trajectories. The motion latent $\mathbf{z}^t$ serves as a bridge between high-level task understanding and low-level control, abstracting the essential motion while filtering out irrelevant visual details.

Our proposed action prediction model is a conditional diffusion policy that generates 6-DoF end-effector trajectories given: (1) the current scene observation, (2) the learned motion latent code, and (3) the robot's proprioceptive state. By conditioning on geometry-aware motion latents rather than raw language instructions or pixels, the policy benefits from motion priors learned across the entire dataset, improving both sample efficiency and generalization.

### 3.3.1 OVERALL PIPELINE

Specifically, we employ a 3D denoising transformer $\epsilon_\theta$ based on Ke et al. (2024) to generate executable trajectories through conditional diffusion. The model iteratively refines noisy action sequences into precise robot motions. Given the current observation $\mathbf{o}^t$, motion latent $\mathbf{z}^t$, and proprioceptive state $\mathbf{c}^t$, the model generates an action chunk $\mathbf{a}^{t:t+h-1}$ over a horizon $h$ via DDPM, where each action $\mathbf{a}^{t+k} = (\mathbf{p}^{t+k}, \mathbf{r}^{t+k}, g^{t+k})$ specifies the end-effector position, rotation, and gripper state.

**Input Tokenization.** The transformer processes four types of tokens (illustrated in Fig. 2): *(1) Motion Latent Tokens*: The discrete motion latent $\mathbf{z}^t = [z_1^t, \ldots, z_{n_s}^t]$ from Sec. 3.2 is embedded via learned embeddings $\mathbf{E}_z \in \mathbb{R}^{n_s \times d_{\text{code}}}$, providing high-level motion guidance. *(2) Trajectory Tokens*: Each noisy action is encoded through an MLP to produce token $\mathbf{t}_{\text{traj}}^k \in \mathbb{R}^{d_{\text{model}}}$. We further concatenate the 3D position $\mathbf{p}^{t+k}$ as positional information to maintain spatial grounding. *(3) Scene Tokens*: We extract visual features $\mathbf{F} \in \mathbb{R}^{H \times W \times 3}$ using a frozen CLIP-ResNet50 (Radford et al., 2021) encoder. Each feature $\mathbf{F}_{ij}$ at spatial location $(i, j)$ is lifted to 3D position $\mathbf{q}_{ij}$ using the depth map and camera intrinsic matrix. This produces $H \times W$ scene tokens $\{(\mathbf{F}_{ij}, \mathbf{q}_{ij})\}$ combining appearance and 3D position. *(4) Proprioception Token*: The robot state $\mathbf{c}^t$ (joint angles, end-effector pose) is encoded as $\mathbf{t}_{\text{prop}} = \text{MLP}(\mathbf{c}^t) + \text{PosEmbed}(\mathbf{p}_{\text{ee}}^t)$, where $\mathbf{p}_{\text{ee}}^t$ is the current end-effector position and PosEmbed denotes the positional embedding.

**Attention Mechanisms.** Then, the transformer employs a two-stage attention strategy to integrate spatial, temporal, and task information: *(1) Self-Attention with 3D Positional Encoding*: We first apply self-attention across all trajectory, scene, and proprioception tokens. To encode spatial relationships, we use the rotary positional embeddings (Su et al., 2021). *(2) Cross-Attention to Motion Latents*: After self-attention, we apply cross-attention from trajectory, scene and proprioception tokens to the motion latent embeddings. This mechanism allows the geometry-aware motion patterns

encoded in $\mathbf{z}^t$ to guide trajectory generation. Unlike conditioning on raw language, the discrete motion latents provide structured, geometrically consistent motion priors based on real-time observation.

Finally, we apply MLPs to predict the noise added to the sequence of 3D translations $\epsilon_{\theta,p}$ and rotations $\epsilon_{\theta,r}$, as well as the gripper state $\hat{g}$, with the final trajectory tokens from the transformer outputs. This progressively refines the action estimate based on scene geometry and learned motion patterns.

### 3.3.2 TRAINING OBJECTIVE AND INFERENCE PROCESS FOR ACTION PREDICTION

**To learn,** we train the denoising transformer $\epsilon_\theta$ to predict the noise added to the ground-truth action chunk. Given a clean action chunk $\mathbf{a}^{t:t+h-1}$ from demonstrations, we sample noise $\epsilon \sim \mathcal{N}(0, I)$ at diffusion step $i$, then create the noisy action chunk: $\mathbf{a}_i = \sqrt{\bar{\alpha}_i} \cdot \mathbf{a}^{t:t+h-1} + \sqrt{1 - \bar{\alpha}_i} \cdot \epsilon$. The model learns to predict $\epsilon$ given the noisy action chunk and conditioning: $\mathcal{L}_\theta = \mathbb{E}_{i,\epsilon} \left[ \|\epsilon - \epsilon_\theta(\mathbf{a}_i, i, \mathbf{o}^t, \mathbf{z}^t, \mathbf{c}^t)\| \right]$. Since actions have components with different scales and properties, we use component-specific losses:

$$\|\epsilon - \epsilon_\theta(\cdot)\| = \lambda_p \|\epsilon_p - \epsilon_{\theta,p}(\cdot)\|_1 + \lambda_r \|\epsilon_r - \epsilon_{\theta,r}(\cdot)\|_1 + \lambda_g \cdot \text{BCE}(g, \hat{g}) \tag{3}$$

where subscripts $p, r, g$ denote position, rotation, and gripper components, respectively. We use L1 loss for continuous values (which is more robust to outliers in trajectory data) and binary cross-entropy (BCE) for the discrete gripper state. The weights $\{\lambda_p, \lambda_r, \lambda_g\}$ are determined with tuning.

**For inference,** we sample a noisy action chunk $\mathbf{a}_N \sim \mathcal{N}(0, I)$ and iteratively denoise it following DDPM (Ho et al., 2020) using the learned model and the current observation's motion latent. The final denoised action chunk contains executable 6-DoF poses and gripper commands that can be directly sent to the robot controller. The geometry-aware motion latent $\mathbf{z}^t$ ensures that the generated trajectory respects both the task semantics and 3D spatial constraints learned from demonstrations.

## 4 EXPERIMENTS

We evaluate *GeoMoLa* through comprehensive experiments designed to answer three key questions: (1) Does 4D spatiotemporal modeling improve manipulation performance compared to 2D/3D baselines? (2) Are the geometry-aware motion latents interpretable and transferable across tasks? (3) Does our approach generalize effectively to real-world scenarios with occlusion and clutter? We conduct experiments on two simulation benchmarks (RLBench (James et al., 2019) and CALVIN (Mees et al., 2021)) and real-world manipulation tasks using the ALOHA robot (Fu et al., 2024), with extensive ablations to validate our design choices.

### 4.1 RLBENCH EVALUATION

**Benchmark description.** RLBench is built on top of the CoppeliaSim simulator (Rohmer et al., 2013), using a Franka Panda robot to interact with the environment. Our model and all baselines are trained to predict the next end-effector keypose rather than the entire trajectory. To execute the predicted keypose, we use RLBench's built-in BiRRT motion planner to generate a feasible trajectory. For evaluation, we select a suite of 10 challenging language-conditioned manipulation tasks, including 166 variations. These variations vary in several types, like position, shape, and color. We use the front-view RGB-D camera as input to comply with practical deployment conditions. Performance is measured by task completion success rate, defined as the proportion of execution trajectories that satisfy the language-specified goal conditions. GNFactor (Ze et al., 2023) and ManiGaussian (Lu et al., 2024) require an additional 19 views for 3D reconstruction during training. All other models receive only the front-view observation as input during inference.

**Baselines.** We compare against three types of methods: *(i) 3D representation-based policies:* Act3D (Gervet et al., 2023) voxelizes the workspace and predicts 3D action maps; 3D Diffuser Actor (Ke et al., 2024) uses 3D feature fields with diffusion-based trajectory generation; GNFactor (Ze et al., 2023) leverages neural radiance fields (Mildenhall et al., 2020) for scene understanding. RVT2 (Goyal et al., 2024) reconstructs scene point cloud for better pose estimation. *(ii) 4D dynamic framework:* ManiGaussian (Lu et al., 2024) employs dynamic 3D Gaussian Splatting (Kerbl et al., 2023) for scene-level spatiotemporal dynamics. *(iii) motion latent methods:* SkillDiffuser (Liang et al., 2023) learns discrete skills from 2D observations without explicit 3D geometric grounding.

Table 1: **Task success rates on RLBench.** *GeoMoLa* achieves the highest average performance (80.1%) and ranks first on 9 out of 10 tasks. Results averaged over 5 seeds.

| Method | Close jar | Open drawer | Sweep to dustpan | Turn tap | Meat off grill | Stack blocks | Slide block | Put in drawer | Drag stick | Push buttons | Avg. |
|---|---|---|---|---|---|---|---|---|---|---|---|
| GNFactor (Ze et al., 2023) | 25.3 | 76.0 | 28.0 | 50.7 | 57.3 | 4.0 | 20.0 | 0.0 | 37.3 | 18.7 | 31.7 |
| ManiGaussian (Lu et al., 2024) | 28.0 | 76.0 | 64.0 | 56.0 | 60.0 | **12.0** | 24.0 | 16.0 | 92.0 | 20.0 | 44.8 |
| Act3D (Gervet et al., 2023) | 52.0 | 84.0 | 80.0 | 64.0 | 66.7 | 0.0 | **100.0** | 54.7 | 86.7 | 64.0 | 65.3 |
| SkillDiffuser (Liang et al., 2023) | 64.2 | 81.0 | 96.6 | 70.6 | 72.1 | 4.0 | 87.0 | 89.2 | 95.6 | 83.8 | 74.4 |
| 3D Diffuser Actor (Ke et al., 2024) | 66.4 | 85.6 | **98.4** | 75.2 | 76.0 | 4.0 | 87.2 | 94.4 | **98.4** | 84.0 | 77.0 |
| RVT2 (Goyal et al., 2024) | 67.0 | 88.2 | **96.2** | 76.0 | 79.4 | **12.0** | 85.1 | 93.8 | **97.8** | 86.2 | 78.1 |
| *GeoMoLa* (Ours) | **69.4** | 85.9 | 98.4 | **81.4** | 79.0 | 12.0 | 90.5 | 95.0 | 98.2 | 92.1 | **80.1** |

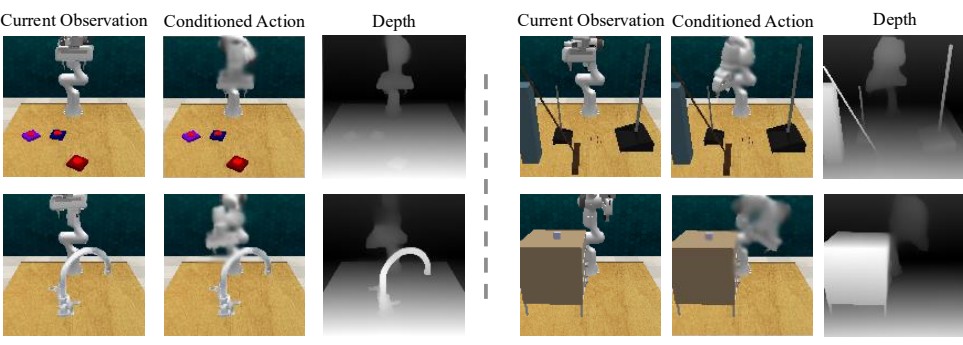

Figure 3: **Cross-scenario motion latent consistency.** Visualization of predicted future observations when conditioning the same latent code on different initial scenes. Left: Latent code $[1, 1, 5, 7]$ consistently produces downward motion. Right: Latent code $[0, 3, 2, 2]$ consistently generates rotational motion. Consistent RGB and depth predictions demonstrate that geometry-aware motion latents capture transferable 3D motion primitives independent of scene content (Please check the synthesized end-effectors).

**Quantitative Results.** As shown in Tab. 1, our method achieves the highest overall performance (80.1%), ranking first on 9 out of 10 tasks. Unlike prior works that condition their action prediction models solely on static 3D feature fields, our approach explicitly models dynamics by learning motion latents from state transitions. While SkillDiffuser also employs latent skill learning, our method further leverages 3D geometric changes along trajectories, enabling more faithful representation of scene structure and, consequently, more precise and reliable action generation.

**Interpretability of Geometry-Aware Motion Latents.** To validate that our learned motion latents encode semantically meaningful motion primitives rather than task-specific behaviors, we conduct cross-scenario generalization experiments. We extract latent codes from successful trajectories and apply them to different scenes, measuring the consistency of the resulting motion patterns.

Fig. 3 demonstrates this cross-scenario transfer capability. When latent code $[1, 1, 5, 7]$ is applied to diverse initial configurations, it consistently generates downward motion in the predicted future observations. Similarly, code $[0, 3, 2, 2]$ reliably produces rotational motion regardless of scene content. This semantic consistency validates that our 4D dynamics pretraining successfully distills reusable motion primitives from the continuous space of robot actions.

**Future Observation Prediction Quality.** Beyond semantic consistency, accurate future prediction is crucial for action planning. We evaluate the quality of predicted observations using both perceptual and geometric metrics. Fig. 4 demonstrates our superior performance compared to another reconstruction-based 4D dynamic method. Despite ManiGaussian's access to 19 additional camera views for the training of dynamic Gaussian Splatting, our pointmap approach produces more accurate predictions with sharper object boundaries. The pointmap representation naturally preserves scene structure during the diffusion process and is much simpler for learning the geometry information, while 3D Gaussian deformation often introduces artifacts at occlusion boundaries due to data scarcity. This prediction fidelity directly impacts action generation – accurate future state prediction enables better trajectory planning, particularly for tasks involving precise object interactions.

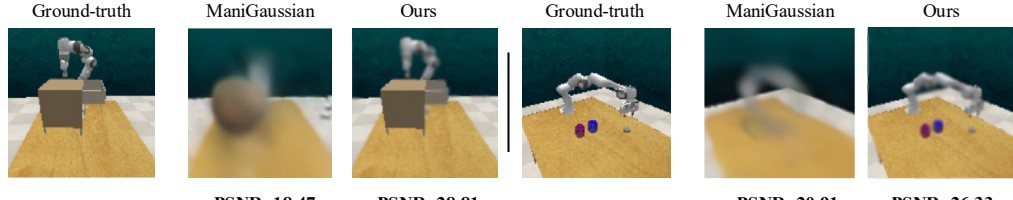

| Ground-truth | ManiGaussian | Ours | Ground-truth | ManiGaussian | Ours |

PSNR: 18.47     PSNR: 28.81        PSNR: 20.91     PSNR: 26.33

Figure 4: **Qualitative comparison of future observation prediction.** ManiGaussian requires 19 additional training views and produces blurred predictions with geometric inconsistencies (see distorted object boundaries). *GeoMoLa* generates sharper, geometrically consistent predictions using only single-view training. We report PSNR for quantitative comparison (higher is better).

Table 2: **Long-horizon task chaining on CALVIN.** Methods are evaluated on completing sequences of 1-5 tasks in the unseen environment D. *GeoMoLa* achieves the longest average task chain (3.58).

| Method | Training Data | Task Completion Rate (%) | | | | | Avg. Length |
|---|---|---|---|---|---|---|---|
| | | 1 Task | 2 Tasks | 3 Tasks | 4 Tasks | 5 Tasks | |
| DP3 (Ze et al., 2024) | Language-annotated | 28.7 | 2.7 | 0.0 | 0.0 | 0.0 | 0.31 |
| GR-1 (Wu et al., 2023) | Language-annotated | 85.4 | 71.2 | 59.6 | 49.7 | 40.1 | 3.06 |
| SuSIE (Black et al., 2023) | All play data | 87.0 | 69.0 | 49.0 | 38.0 | 26.0 | 2.69 |
| RVT2 (Goyal et al., 2024) | Language-annotated | 90.4 | 76.8 | 61.6 | 50.2 | 39.8 | 3.23 |
| 3D Diffuser Actor (Ke et al., 2024) | Language-annotated | 93.8 | 80.3 | 66.2 | 53.3 | 41.2 | 3.35 |
| Clover (Bu et al., 2024) | Language-annotated | **96.0** | 83.5 | 70.8 | 57.5 | 45.4 | 3.53 |
| *GeoMoLa* (Ours) | Language-annotated | 95.4 | **84.0** | **72.4** | **59.0** | **46.4** | **3.58** |

## 4.2 CALVIN EVALUATION

**Benchmark Description.** CALVIN (Mees et al., 2021) evaluates long-horizon task execution in PyBullet, requiring sequential completion of language-specified sub-tasks. It includes 34 distinct tasks across 4 environments (A, B, C, D) with varying textures and object positions. Each environment contains a Franka Emika Panda robot, desk, sliding door, drawer, LED button, light switch, and colored blocks. CALVIN provides 24 hours of teleoperated unstructured play data, 35% of which are annotated with language descriptions (18k trajectory videos). Each instruction chain includes five language instructions that need to be executed sequentially. We evaluate on the challenging zero-shot generalization setting: training on environments A, B, C and testing on the unseen environment D.

**Baselines.** We compare against: (i) *3D-based methods*: DP3 (Ze et al., 2024) encodes RGB-D into 3D features for diffusion trajectory prediction (we add language conditioning following (Ke et al., 2024)); 3D Diffuser Actor and RVT2 that previously mentioned. (ii) *Video-pretrained models*: GR-1 (Wu et al., 2023) leverages large-scale internet video pretraining; SuSIE (Black et al., 2023) uses all available play data including unannotated sequences. (iii) *Closed-loop methods*: Clover (Bu et al., 2024) encodes RGB-D inputs, introducing a closed-loop error correction mechanism to achieve better video diffusion policy.

**Results Analysis.** Tab. 2 shows *GeoMoLa* achieves the highest average task sequence length of 3.58, completing 46.4% of 5-task chains compared to 41.2% for 3D Diffuser Actor. The performance gap widens with sequence length – while Clover's closed-loop correction provides only a 0.6% advantage on completion of the first subtask, *GeoMoLa* outperforms other methods when completing the entire long-horizon subtask sequences. This trend validates our hypothesis: explicit 4D dynamics modeling reduces compounding errors that accumulate over long horizons. While RVT2 utilizes RGB-D input, it obtains supervision only from ground-truth poses. Our results demonstrate that learning motion patterns enables more accurate action prediction without requiring additional views.. The stark failure of DP3 (0.31 average length) despite using 3D representations highlights that static geometric features alone are insufficient for sequential manipulation. GR-1's internet-scale pretraining provides strong single-task performance (85.4%) but degrades rapidly in multi-task scenarios, suggesting that generic video understanding does not transfer directly to precise robotic control.

Table 3: **Impact of 4D dynamics learning.**

| Method Variant | CALVIN (Task Completion %) | | | | | | RLBench |
|---|---|---|---|---|---|---|---|
| | 1 Task | 2 Tasks | 3 Tasks | 4 Tasks | 5 Tasks | Avg. Len | Success Rate |
| *GeoMoLa* w/o Pointmap | 94.4 | 81.0 | 67.6 | 54.9 | 43.2 | 3.38 | 78.0% |
| *GeoMoLa* w/o RGB | 95.0 | 82.3 | 69.6 | 55.2 | 45.0 | 3.50 | 79.4% |
| *GeoMoLa* (Full) | **95.4** | **84.0** | **72.4** | **59.0** | **46.4** | **3.58** | **80.1%** |

### 4.3 ABLATION STUDIES

To validate our architectural choices, we conduct ablation studies examining the contribution of each modality in the 4D dynamics learning phase. We design two baselines that pretrain the motion latent space using only a single modality – either 2D image prediction or pointmap prediction – by removing the pointmap or RGB prediction branch from our framework, respectively.

**Importance of Geometric vs. Appearance Modeling.** Tab. 3 reveals a critical insight: removing pointmap prediction causes a substantial performance drop (CALVIN: -0.20 avg. length; RLBench: -2.1%), while removing RGB prediction has minimal impact (-0.08 avg. length; -0.7%). This asymmetry demonstrates that explicit 3D geometry is fundamental for learning transferable manipulation primitives, while appearance primarily provides auxiliary context.

**The role of geometry in different motion types.** By analysing the task-specific results on RLBench, we find that in rotation-heavy tasks (e.g., "sweep to dustpan", "open drawer"), removing point map prediction reduced success rates by nearly 9% on average, while in translation-heavy tasks (e.g., "close jar", "push buttons"), the drop was less than 1%. This suggests that 3D geometry is especially important for tasks involving fine rotational control and complex object interactions, aligning with our motivation that geometric awareness supports more precise gripper state estimation and interaction reasoning. Thus, the key value of our approach lies in providing consistent robustness in geometrically complex settings, rather than a uniform boost across all tasks.

### 4.4 REAL-WORLD VALIDATION

**Experimental Setup.** We collected real-world demonstration data using the ALOHA robot platform. RGB-D observations are captured via an Intel RealSense camera at 640×480 resolution from a front view and subsequently downsampled to 256×256 for processing. During inference, target gripper poses are executed using the MoveIt package in ROS (Coleman et al., 2014).

**Data Collection and Training.** We consider six distinct tasks, with 20 demonstration trajectories recorded for each task. To ensure diversity within each task, variations in object quantities, positions, colors, and other attributes were intentionally introduced across different trajectories. The collected demonstration data were subsequently used to train our model. During the testing phase, the model was evaluated on the same set of tasks under zero-shot transfer conditions, where object configurations, spatial layouts, and visual properties such as color were systematically altered to assess generalization beyond the training demonstrations. We evaluated 10 episodes for each task and reported the success rate. Models are trained from scratch on this limited data to evaluate sample efficiency in real-world settings. Detailed task information and visualization of our settings could be found in Appendix. C.

Table 4: **Real-world manipulation performance.** Success rates over 10 trials per task. *GeoMoLa* shows consistent improvements, particularly in cluttered scenarios (Clean cup) and precise manipulation (Stack cubes). † indicates tasks with significant occlusion during execution.

| Method | Clean cup† | Stack cups | Put cups on shelf | Stack cubes | Place dish | Place cube | Average |
|---|---|---|---|---|---|---|---|
| SkillDiffuser (Liang et al., 2023) | 30.0% | 0.0% | 20.0% | 20.0% | 20.0% | 40.0% | 21.7% |
| 3D Diffuser Actor (Ke et al., 2024) | 50.0% | 10.0% | **40.0%** | 20.0% | 40.0% | 80.0% | 40.0% |
| GeoMoLa w/o Pointmap | 20.0% | 10.0% | 30.0% | 20.0% | **50.0%** | 80.0% | 35.0% |
| *GeoMoLa* (Ours) | **60.0%** | **30.0%** | **40.0%** | **50.0%** | **50.0%** | **90.0%** | **53.3%** |

**Results and Analysis.** Tab. 4 shows *GeoMoLa* achieving a 53.3% average success rate, a 13.3% improvement over 3D Diffuser Actor. In particular, we observe consistent gains in tasks such as "Clean cup", which involves significant occlusion, as well as "Stack cups" and "Stack cubes", which

require precise manipulation. These improvements highlight the benefits of our approach in modeling dynamics and scene geometry, enabling more accurate action prediction. More detailed visualization discussion can be found in Fig. 1 and the Appendix C. The results demonstrate the effectiveness of our method for real-world robotic manipulation in diverse and previously unseen settings.

## 5 CONCLUSION

This work demonstrates that motion latent learning for robotic manipulation benefits significantly from grounding in four-dimensional geometric transformations rather than visual sequences. Our ablation studies provide quantitative evidence: removing geometric prediction degrades motion latent quality substantially while visual prediction contributes minimally to performance. The success with limited real-world demonstrations indicates that geometry-aware latent representations naturally capture manipulation-relevant motion primitives without requiring extensive datasets. Future work could extend this motion latent framework to deformable objects and investigate hierarchical planning where high-level policies compose learned geometric primitives. Our results suggest that effective motion latent representations for robotics should encode how objects move through three-dimensional space over time rather than how they appear visually, providing another perspective on learning reusable manipulation skills from unlabeled demonstrations.

## 6 ETHICS STATEMENT

This work strictly adheres to established research ethics guidelines. Our research focuses on developing geometry-aware motion learning for robotic manipulation and does not involve human subjects, animal experiments, or raise concerns related to privacy or security. All experiments were conducted in simulation environments using synthetic data and self-collected demonstrations from consenting researchers, with no personal data, biometric information, or sensitive content involved. The geometry-aware motion learning framework is designed for research purposes in controlled robotic manipulation tasks and presents minimal risks of misuse. We have carefully considered the broader impacts of our work and believe it contributes positively to the advancement of reliable and sample-efficient robotic systems. The improved generalization capabilities and reduced data requirements demonstrated by our method could benefit applications in assistive robotics, healthcare, elder care, and accessibility when properly validated in real-world settings. We acknowledge that advances in robotic automation may have economic implications for certain job categories. However, our work focuses specifically on benign manipulation tasks such as grasping, stacking, and placing objects, with the goal of enhancing human-robot collaboration rather than replacement. The framework is designed to augment human capabilities in controlled environments rather than substitute for human judgment and creativity. While any advancement in robotic capabilities could potentially be misapplied, our research addresses fundamental manipulation learning without developing applications for harmful purposes. All authors have thoroughly reviewed this work and acknowledge compliance with research ethics standards.

## 7 REPRODUCIBILITY STATEMENT

We have taken extensive measures to ensure the reproducibility of our work on the GeoMoLa framework. The complete architectural details of our geometry-aware latent action learning pipeline and diffusion-based action prediction model are described in Sec. 3 and Appendix D. All experimental configurations, including task specifications for RLBench and CALVIN benchmarks, batch size, learning rate, training epochs, and diffusion model parameters are detailed in Sec. 4 and Appendix B, C, D. The geometry-aware encoding formulation, latent action space construction, and diffusion-based prediction procedure are precisely specified with corresponding equations and algorithms provided in Sec. 3 and Appendix D. Our experimental evaluation builds upon established publicly available benchmarks (RLBench and CALVIN) using standard evaluation protocols, with all success metrics and evaluation criteria clearly defined for direct comparability with existing work. Implementation details include the geometry encoder architecture, latent space dimensionality, diffusion model configuration, and specific hyperparameters for all baseline methods documented in Sec. 3 and Appendix D. Our ablation studies examining the contributions of each component are systematically presented with quantitative results. For real-world validation, we provide detailed collection procedures for our demonstration dataset, including robot setup, task specifications, and data preprocessing steps. Upon publication, we will make available the complete implementation of GeoMoLa, including trained models, configuration files, example usage scripts, and our collected demonstration dataset with comprehensive documentation to facilitate reproduction of all reported results and enable extension of our work.

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

## A   THE USE OF LARGE LANGUAGE MODELS

This work utilized large language models as supplementary tools to enhance writing quality, including improving clarity, maintaining consistency between sections, and refining adherence to academic writing conventions. The core research concepts, methodological approaches, and findings represent original contributions by the authors.

## B   SIMULATION EXPERIMENTS

### B.1   DATASET COMPOSITION

**RLBench.**   RLBench is a comprehensive and large-scale benchmark and learning environment designed to advance research in vision-guided robotic manipulation. The platform is tailored to support a variety of research areas, including reinforcement learning, imitation learning, multi-task learning, and few-shot learning.

In our specific experimental setup, we utilize a subset of 10 manipulation tasks from the RLBench environment to evaluate the multi-task capabilities of our agents. These tasks are selected to cover a diverse range of challenges, involving different objects, objectives, and required skills. The variations within each task are designed to test an agent's ability to understand and adapt to changes in color, placement, size, and object category.

Tab. 5 shows the details of the composition of the RLBench dataset.

| Task | Variation Type | # of Variations | Avg. Keyframes | Language Description Example |
|---|---|---|---|---|
| close jar | color | 20 | 6.0 | "close the — jar" |
| meat off grill | category | 2 | 5.0 | "take the — off the grill" |
| open drawer | placement | 3 | 3.0 | "open the — drawer" |
| sweep to dustpan | size | 2 | 4.6 | "sweep dirt to the — dustpan" |
| turn tap | placement | 2 | 2.0 | "turn — tap" |
| slide block | color | 4 | 4.7 | "slide the block to — target" |
| put in drawer | placement | 3 | 12.0 | "put the item in the — drawer" |
| drag stick | color | 20 | 6.0 | "use the stick to drag the cube onto the — — target" |
| push buttons | color | 50 | 3.8 | "push the — button, [then the — button]" |
| stack blocks | color, count | 60 | 14.6 | "stack — — blocks" |

Table 5: Dataset composition of 10 manipulation tasks in RLBench James et al. (2019).

**CALVIN.**   A key evaluation protocol within CALVIN is the "ABC→D" setup, which is specifically designed to test an agent's ability to generalize to a new, unseen environment. This setup is considered one of the most challenging evaluations in the benchmark.

The visualization of the Calvin setting is demonstrated in Fig. 5. This zero-shot generalization task, where the agent must apply learned skills to a completely new setting, is crucial for assessing the robustness and adaptability of the control policy. The ABC→D setup measures how well a policy can transfer its understanding of language and manipulation to an unfamiliar setting.

### B.2   ADDITIONAL VISUALIZATION OF FUTURE OBSERVATIONS PREDICTION

Besides the visualization shown in Fig. 4 on RLBench, we also provide the future observations prediction results on Calvin. Fig. 6 illustrates the generated future observations conditioned on the current scene and the inferred latent action $\mathbf{z}^t$ for the "Open the drawer" task. Our model produces highly accurate and temporally consistent predictions in both RGB and depth modalities. The generated frames faithfully capture the geometric displacement of the drawer and the manipulator trajectory over multiple timesteps, showing smooth and physically plausible motion progression. Importantly, the predicted depth maps remain well aligned with the RGB predictions, indicating that the model preserves 3D scene structure rather than merely hallucinating pixel-level appearance. This coherence highlights that the latent action representation encodes the causal effect of actions in 3D space, enabling the diffusion model to generate consistent future trajectories.

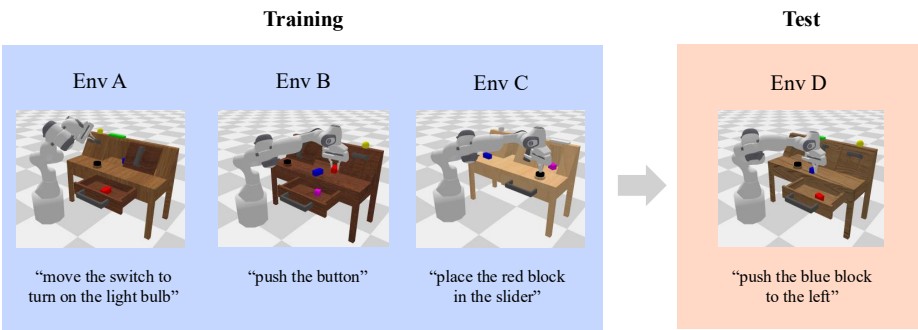

Figure 5: **llustration of the four different environments in CALVIN (Mees et al., 2021).**

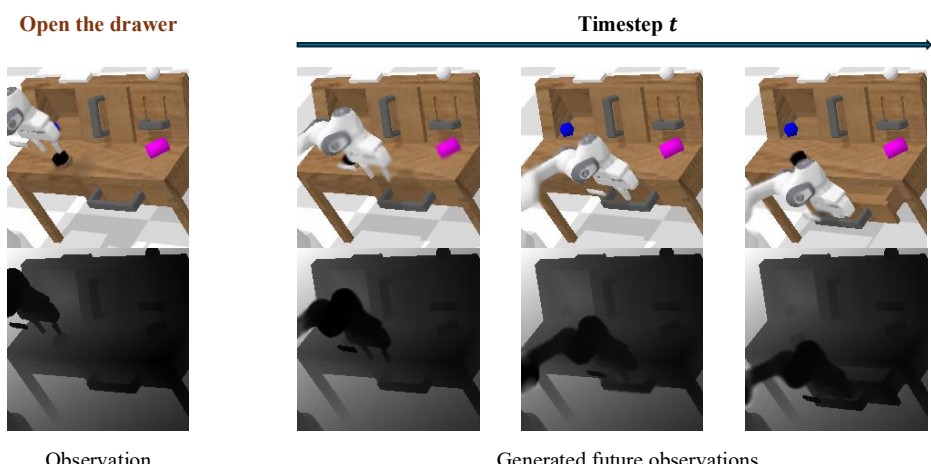

Figure 6: **Future observation prediction on Calvin (Mees et al., 2021).**

## C    REAL-WORLD EXPERIMENTS

### C.1    DATASET COMPOSITION

| Task | Variation Type | Language Description Example |
|---|---|---|
| Place dish | color, count | "Place the — dish on the — tablecloth" |
| Clean cups | color, count, placement | "Put the — cup into the — basket" |
| Stack cups | color, count, placement | "Stack the — cup on the — cup" |
| Stack cubes | color, count | "Stack the — cubes" |
| Put cups on shelf | placement | "Put the — cup on the shelf next to the — cup" |
| Place cube | color, count | "Place the — block on the — plate" |

Table 6: Dataset composition of 6 manipulation tasks in real robot experiments.

We evaluate our approach on six real-world manipulation tasks, each incorporating controlled variations in object placement, color, and count. This design introduces perceptual and spatial diversity, challenging the agent to generalize across visually distinct yet semantically similar scenarios. The visualization of our settings is shown in Fig. 7.

### C.2    MORE EXPERIMENTAL RESULTS

**Real-World Experiments.**    We deploy our trained policy on a real Franka Panda manipulator and evaluate it on six long-horizon tabletop manipulation tasks: (1) "Put the blue cup on the shelf next to

Figure 7: **Visualization of the real-world experiment setup.**

the green cup", (2) "Stack the yellow cup on the green cup", (3) "Place the green cube on the pink plate", (4) "Stack the orange cubes", (5) "Place the blue dish on the blue tablecloth", and (6) "Put the yellow cup into the white basket". As shown in Fig. 8, our policy successfully completes all six tasks with smooth and collision-free trajectories, demonstrating strong sim-to-real transfer. The robot consistently executes precise grasps, object placements, and stacking behaviors, even in cluttered and visually diverse scenes. These results confirm that the learned latent actions generalize to real-world execution and maintain their semantic meaning outside of the simulation domain.

**Case study of precise action control.** Fig. 9 presents a qualitative comparison of action execution in a real-world cluttered scene, highlighting the advantage of our 4D modeling approach over a 3D static baseline. The top row (3D Diffuser Actor) demonstrates failure: although the robot successfully grasps and lifts the target cup, it misaligns during placement due to a lack of temporal dynamics and spatial reasoning — resulting in an unstable or incorrect stack. In contrast, our method (bottom row) leverages 4D spatio-temporal modeling to predict not only where but when and how to act, enabling precise control throughout the motion trajectory. As shown, our agent successfully stacks the cup with stable alignment, even under visual occlusion and object clutter. This illustrates that 4D-aware policy learning is critical for achieving reliable, fine-grained manipulation in dynamic physical environments — a capability absent in purely 3D state-based models.

# D  IMPLEMENTATION DETAILS

## D.1  LEARNING GEOMETRY-AWARE LATENT ACTIONS.

To learn geometry-aware latent actions, we employ two different branches of diffusion models to predict the future observations.

Given RGB-D observation $\mathbf{o}^t$, we convert it to a pointmap:

$$\mathbf{P}^t = \text{BackProject}(\mathbf{o}^t) \in \mathbb{R}^{H \times W \times 3}.$$

**Latent Encoding.** Pointmaps are encoded by a 3D-aware VAE:

$$\mathbf{h}_{\text{pm}}^{t'} = \psi^{\text{enc}}(\mathbf{P}^{t'}), \quad \hat{\mathbf{P}}^{t'} = \psi^{\text{dec}}(\hat{\mathbf{h}}_{\text{pm}}^{t'}).$$

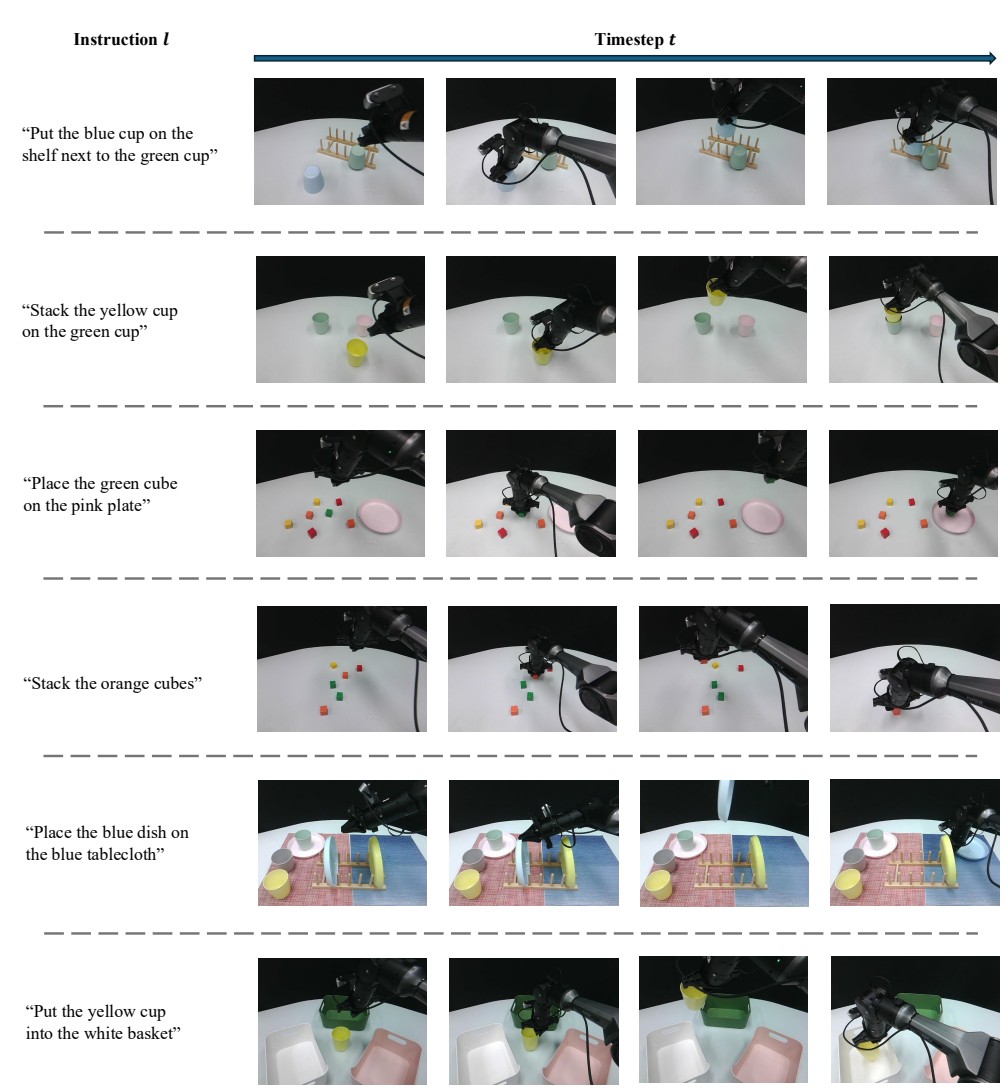

Figure 8: **More qualitative results on real-world experiments.**

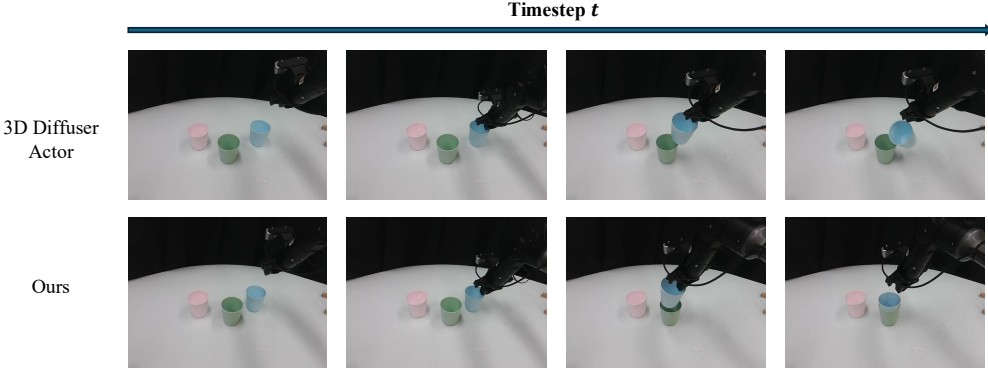

Figure 9: **Qualitative comparison in the real-world cluttered scene.**

The pointmap VAE is initialized by the RGB VAE. Before the latent action learning, we first finetune it on the video sequences of demonstration for a few epochs.

**Latent Action-Conditioned Future Prediction.** We generate future latents with a conditional diffusion model adapted from DDPM Ho et al. (2020) with motion latent $z^t$:

$$\hat{\mathbf{h}}_{\text{pm}}^{t+1:t+w} = \psi^{\text{diff}}\big(\mathbf{h}_{\text{pm}}^{t-w+1:t}, \mathbf{z}^t; \phi\big) \,.$$

At each diffusion step $k$, we add Gaussian noise:

$$\mathbf{h}_k^{\text{pm}} = \sqrt{\alpha_k} \cdot \mathbf{h}_{\text{pm}}^{t+1:t+w} + \sqrt{1 - \alpha_k} \cdot \epsilon, \quad \epsilon \sim \mathcal{N}(0, I),$$

and minimize the denoising objective:

$$\mathcal{L}_{\text{diff}}^{\text{pm}} = \mathbb{E}_{k,\epsilon}\big[\|\epsilon - \epsilon_\phi(\mathbf{h}_k^{\text{pm}}, k, \mathbf{h}_{\text{pm}}^{t-w+1:t}, \mathbf{z}^t)\|^2\big] \,.$$

**Joint RGB Prediction.** An RGB branch predicts future appearance using the same architecture and conditioning:

$$\mathcal{L}_{\text{diff}}^{\text{rgb}} = \mathbb{E}_{k,\epsilon}\left[\|\epsilon - \epsilon_\phi^{\text{rgb}}(\mathbf{h}_k^{\text{rgb}}, k, \mathbf{h}_{\text{rgb}}^{t-w+1:t}, \mathbf{z}^t)\|^2\right] \,.$$

**Vector-Quantized Latent Regularization.** Latent actions are discretized using VQ-VAE loss:

$$\mathcal{L}_{\text{vq}} = \|\text{sg}[\mathbf{f}^t] - \mathbf{c}\|_2^2 + \beta\|\mathbf{f}^t - \text{sg}[\mathbf{c}]\|_2^2.$$

**Overall Objective.** The total training loss is:

$$\mathcal{L}_{\text{total}} = \mathcal{L}_{\text{diff}}^{\text{pm}} + \mathcal{L}_{\text{diff}}^{\text{rgb}} + \mathcal{L}_{\text{vq}}.$$

Note that both two VAEs are all frozen during the training process.

## D.2 ACTION PREDICTION WITH GEOMETRY-AWARE LATENT ACTIONS.

The 3D denoising transformer $\epsilon_\theta$ is a multi-layer conditional diffusion model that iteratively refines noisy action sequences into executable robot trajectories. The model takes four types of input tokens: trajectory tokens, scene tokens, proprioception tokens, and latent action tokens.

**Token Embeddings.** We embed each noisy action $\mathbf{a}^{t+k}$ using a MLP, producing a $d_{\text{model}} = 256$-dimensional trajectory token. Scene features are extracted using a frozen CLIP-ResNet50 encoder and lifted to 3D with depth information and camera intrinsics. The proprioceptive state $\mathbf{c}^t$ is encoded with an MLP and added with the positional embedding of the end-effector pose.

**Transformer Backbone.** The denoising network contains: (i) a multi-head self-attention layer over all trajectory, scene, and proprioception tokens with rotary 3D positional encodings; (ii) a cross-attention layer that attends from all tokens to the latent action embeddings $z_t$ to inject high-level motion priors; In order to decrease the computational requirements, we subsample a number of visual tokens using Farthest Point Sampling (FPS). The sampled visual tokens, proprioception tokens, and noisy position/rotation tokens attend to each other.

**Output Heads.** The final transformer layer outputs refined trajectory tokens, which are decoded using two independent MLP heads: One predicts the noise of position and 6D rotation, and the other predicts the gripper open/close state.

**3D relative attention.** We formulate the detailed attention as follows:

$$a_{q,k} \propto x_q^\top M(p_q - p_k) x_k$$

- $a_{q,k}$: attention weight between query token $q$ and key token $k$
- $x_q$: feature vector of the query token
- $x_k$: feature vector of the key token
- $p_q$: 3D position of the query token
- $p_k$: 3D position of the key token
- $M(p_q - p_k)$: matrix-valued function that depends only on the relative 3D position between query and key

|  | RLBench | CALVIN |
|---|---|---|
| **Transformer** | | |
| image_size | 256 | 200 |
| embedding_dim | 120 | 192 |
| camera_views | 1 | 2 |
| FPS : % of sampled tokens | 20% | 33% |
| diffusion_timestep | 100 | 25 |
| **Latent action** | | |
| Patch size | 16 | 16 |
| Hidden size | 768 | 768 |
| Codebook size | 64 | 64 |
| Codebook dim | 32 | 32 |
| **Training** | | |
| batch_size | 240 | 5400 |
| learning_rate | $1e^{-4}$ | $3e^{-4}$ |
| weight_decay | $5e^{-4}$ | $5e^{-3}$ |
| total_epochs | $1.6e^4$ | 90 |
| optimizer | Adam | Adam |

Table 7: Comparison of configurations between RLBench and CALVIN.

## D.3 HYPER-PARAMETERS

The summary of used hyper-parameters for training/evaluating our model is described in Tab. 7.