# OpenReview forum: "GeoMoLa: Geometry-Aware Motion Latents for Learning Robust Manipulation Policies"
_ICLR.cc/2026/Conference — Submitted to ICLR 2026_

### Official Review · Reviewer_voeQ · 2025-10-27

**Soundness:** 2
**Presentation:** 2
**Contribution:** 2
**Rating:** 4
**Confidence:** 3

**Summary:**

The paper proposes a self-supervised way of learning discrete action codes through predicting how point clouds evolve during manipulation rather than reconstructing visual observations. The paper argues that the spatial geometry changing through time is important and useful to spatial understanding. Experiments on different benchmarks show superior performances. The paper claims they are the first framework that models robot manipulation as continuous four-dimensional process.

**Strengths:**

* The proposed method only uses single-view RGB-D inputs, while still achieving competitive performance. This is important in real-world deployment of robots.
* The intuition behind the proposed method makes sense. The learned results show good interpretability also hints the effectiveness of the method.

**Weaknesses:**

* The methodology section is quite hard to follow. There is few logical connection between each subsection. The writing mostly consists of plain descriptions of models, without many explanation, which makes the framework hard to understand.
* The logical connection of Figure 2 is unclear. What is the relation between (a) and (b)? Do they share any module?
* It is hard to see how the four-dimensional geometry changing is imposed in the training objective. It looks like from line 223 the constraint is imposed through predicting future latent point map features. And the latent features are obtained through a finetuned diffusion model described in Appendix D. The point map VAE is initialized by an RGB VAE, which does not make too much sense to me since 3d coordinates are different modalities than RGB. And Appendix D has some undefined variables such as z^t.

Overall the writing of the paper seems a big issue for me. It is quite confusing so that I find it hard to evaluate the correctness of the architecture. Although the paper has good intuition and seemingly good results, I would rate a borderline rejection.

**Questions:**

Which specific loss function imposes the constraint that "encode actual physical motion rather than appearance patterns"?

---

> ### Author Response · Authors · 2025-11-25
> **Repsond to the reviewer (part 1)**
>
> We sincerely thank the reviewer for the acknowledgment of our intuition and the practical significance of real-world employment. We also thank you for the careful reading and constructive suggestions, which have been invaluable in helping us improve the clarity and rigor of our work. Below, we address each of the concerns raised in the review.
>
> > **Weakness 1** – Methodology Clarity and Logical Flow
>
> We appreciate the feedback regarding the logical flow and clarity of the methodology section. In response, we have revised the method description to strengthen the connections between subsections. We have tried to add some explanatory sentences at the beginning of each subsection in our original version (e.g., Lines 130, 206, 236), and now we supply more relevant phrases (highlighted in blue) to better guide the reader through the overall framework in our revision. We apologize for any remaining confusion and have further refined the revised method section (highlighted in blue) to improve readability and logical coherence.
>
> > **Weakness 2**  – Relationship Between Figure 2(a) and 2(b)
>
> Thank you for pointing out the unclear relationship in Figure 2. Indeed, both (a) and (b) share the same Vision-Language Model (VLM) and the following latent codebook. The process in (a) learns motion-aware latents by predicting future observations, which are then used in (b) to condition the action generation transformer. To clarify this, we have updated Figure 2 and its caption to explicitly illustrate the shared components and the flow of information between the two parts.
>
> > **Weakness 3** – 4D Geometry Constraint and VAE Initialization
>
> As the reviewer noted, the 4D geometry-changing objective is implemented via future point map prediction. By predicting how the 3D geometry evolves over time, the model is forced to capture object motion and interaction dynamics, which are inherently geometric and not tied to appearance. **The specific loss function that enforces this is the denoising objective of the latent diffusion model, described in Line 226**. This objective ensures that the model captures temporal geometric evolution rather than static appearance.
>
> Regarding the initialization of the PointMap VAE with an RGB VAE: **this is an empirical design choice**, inspired by prior work ([1]), which showed that such initialization can aid convergence. While the performance gains are modest (e.g., +0.2% SR on RLBench, +0.03 Avg. Len on CALVIN), it provides a stable starting point for training. We have also clarified the definition of $z^t$ in Appendix D to resolve any ambiguity.

---

> ### Author Response · Authors · 2025-11-25
> **Respond to the reviewre (part 2)**
>
> Additional ablation studies are provided to evaluate the effectiveness of the architectural design, in order to better answer the proposed weakness and question:
>
>
> **Table R1. Task success rate on real-world experiments.**
> | Method | Clean cup | Stack cups | Put cups on shelf | Stack cubes | Place dish | Place cube | Average |
> |---|---|---|---|---|---|---|---|
> | GeoMoLa w/o Pointmap| 20.0% | 10.0% | 30.0% | 20.0% |**50.0%** | 80.0% | 35.0% |
> | GeoMoLa (Ours) | **60.0%** | **30.0%** | **40.0%** | **50.0%** | **50.0%** |**90.0%** | **53.3%** |
>
>
> The above table compares the results with and without the pointmap diffusion module in real-world experiments. As shown, removing pointmap prediction leads to a notable performance drop—for instance, a 40% absolute improvement in heavily occluded tasks such as ***clean cups***. In contrast, the gap is smaller in less cluttered scenarios like ***place cube*** and ***place dish***. These results confirm that 3D geometry learning is particularly beneficial in complex, occlusion-rich environments. Scene visualizations are provided in Figure 1 (***clean cups***) and Figure 7 in Appendix C.
>
>
> We further analyze the role of geometric modeling in different types of motions by evaluating four representative tasks in RLBench, categorized by their dominant motion type:
>
> **Table R2. Success rates on four tasks on RLBench. ("rot." indicates rotation-heavy tasks, and "trans." indicates translation-heavy task)**
> | Method | Close jar (trans.) | Push buttons (trans.) | Open drawer (rot.) | Sweep to dustpan (rot.) |
> |---|---|---|---|---|
> | GeoMoLa w/o Pointmap| 69.0 | 91.6| 76.0 | 90.2 |
> | GeoMoLa | **69.4** | **92.1** | **85.9** | **98.4** |
>
>
> In rotation-heavy tasks (e.g., ***sweep to dustpan, open drawer***), the absence of point cloud prediction reduces the success rate by nearly 9% on average, whereas the impact is minimal (<1%) in translation-heavy tasks (e.g., ***close jar***, ***push buttons***). This suggests that 3D geometry is especially critical in tasks requiring fine rotational control and complex object interactions, supporting our hypothesis that geometric awareness facilitates more precise gripper state estimation and interaction reasoning. **The ablation study in Tab. 3 that evaluates the overall results on both benchmarks further validates the soundness of our design.**
>
> **Regarding visualization and interpretability**, we provide visualizations in Fig. 3 demonstrating that our motion latent can generate semantically meaningful 4D future observations (RGB and depth) across diverse scenarios. This indicates that our model architecture successfully guides the latent features to capture scene geometry and motion dynamics.
>
>
> ## **Summary**
> We deeply appreciate the reviewer’s emphasis on writing quality, which has motivated us to significantly revise the paper for better clarity and understanding. We hope the revised version (**including writing, newly added results, and modified Fig.2**) addresses your concerns and provides a clearer presentation of our framework. We look forward to your further feedback.
>
> **References**
>
> [1] Liu, Zeyi, et al. "Geometry-aware 4D Video Generation for Robot Manipulation." arXiv preprint arXiv:2507.01099 (2025).

---

### Official Review · Reviewer_ViXr · 2025-11-02

**Soundness:** 3
**Presentation:** 2
**Contribution:** 2
**Rating:** 2
**Confidence:** 5

**Summary:**

The paper proposes GeoMoLa. It learns discrete motion codes by predicting future 3D geometry (pointmaps) and RGB from current RGB‑D observations and language.

A VQ‑VAE discretizes vision‑language features into codes; a pointmap/RGB latent diffusion model is trained to forecast future observations conditioned on those codes; and a 3D denoising transformer uses the codes to generate 6‑DoF action chunks.

Experiments on RLBench, CALVIN, and six real‑robot tasks show consistent gains over 2D/3D baselines, with ablations indicating geometry prediction is the main driver of performance.

**Strengths:**

1. Method is well motivated: VQ‑VAE for discrete codes; pointmap diffusion for future geometry; 3D‑aware transformer with relative 3D attention and cross‑attention to latents for action generation.

2. Good experimentation: solid benchmarks and real‑world evaluation with low demo counts, plus clean ablation identifying geometry prediction as the key contributor.

**Weaknesses:**

1. The main contribution to me is tying discrete latents to future 3D geometry prediction and then using them to condition 3D‑aware action diffusion. Might better tone down “first 4D” phrasing and draw distinction vs dynamic Gaussian / NeRF‑style approaches.

2. Baselines are not strong enough. For example, in RLBench experiments, RVT2 is not included. It gets 100% on close jar and 80% on stack block.

2. Besides, the presentation is not very clear. Latent motion / latent action is not consistent and thus confusing to readers. For example,  fig2 has (a) Geometry-Aware Latent Action Learning, and (b) Latent-Conditioned Action Generation. If I get it right, the latent action is to motion latents in the title. But it could mean latent embedding learned for robot action space. Thereby it is unclear to the latent action learning actually until reading much more in depth.

**Questions:**

What is the motivation of deriving latent action from rib and language using minigpt?
Would it be more natural to have depth / point map as input to latent action as well? considering they are assumed available in both training and inference

---

> ### Author Response · Authors · 2025-11-23
> **Respond to the reviewer (part 1)**
>
> We sincerely thank the reviewer for their thoughtful review and for recognizing the strengths of our work. We are particularly grateful for the acknowledgment of:
>
> 1. Our well-motivated methodology integrating VQ-VAE, pointmap diffusion, and a 3D-aware transformer for action generation.
>
> 2. The thorough experimentation across multiple benchmarks, including real-world tasks with low demonstration counts, and the clean ablation studies that identify geometry prediction as the key performance contributor.
>
> Below, we provide detailed responses to the specific concerns raised regarding presentation clarity and baseline comparisons.
>
> > **Weakness 1**: "First 4D" Phrasing
>
> We appreciate the reviewer's insightful comment regarding our "first 4D" phrasing and the distinction from related approaches. We will gladly revise the manuscript to provide clearer motivation and avoid potential confusion.
>
> Our initial phrasing stemmed from observing that while latent action models are widely used in interactive domains like robotics [1] and game AI [2], they have been primarily applied to 2D video data. We argue that for physical robot manipulation, actions are inherently tied to changes in 3D geometry. Our core contribution thus lies in the **novel integration of a 4D geometric prediction objective (3D space + time) into latent action learning specifically for robotics**, which, to our knowledge, represents a new direction in this domain.
>
> We will also clearly distinguish our work from dynamic Gaussian/NeRF-style approaches. The key difference lies in both the objective and the application domain. While those methods excel at high-fidelity dynamic scene reconstruction for novel-view synthesis [3], GeoMoLa is specifically designed for robot policy learning. Our use of 4D geometry prediction serves as a self-supervised objective that forces latent codes to capture physical motion patterns, which are then directly leveraged for action generation.
>
> We agree that refining the "first" claim and adding this clarification will make our contribution more precise. We will modify the relevant explanations in the abstract and introduction to better position our work within the landscape of 4D generation and robotics research.
>
> > **Weakness 2**: Baseline Comparisons
>
> We thank the reviewer for raising this important point about baseline comparisons. Our experiments were conducted under a single-view setting, which represents a common and practical configuration for real-world robot systems. We note that the real-world experiments in RVT2 also employ a single-view setup. For simulation environments, CALVIN also only has two cameras (one static + one mounted).
>
> In RLBench, while RVT2 utilizes four different views to generate point clouds, our model and selected baselines use only front-view input (as mentioned in Line 309). To ensure fair comparison, we conducted additional experiments under this single-view configuration and observed that our approach consistently outperforms RVT2 on both RLBench and CALVIN benchmarks:
>
>
> **Table R1. Success rate on RLBench**
>
> | Method | Close jar | Open drawer | Sweep to dustpan | Turn tap | Meat off grill | Stack blocks | Slide block | Put in drawer | Drag stick | Push buttons | Avg. |
> |---|---|---|---|---|---|---|---|---|---|---|---|
> | **RVT2** (Goyal et al., 2024) | 67.0 | **88.2** | 96.2 | 76.0 | **79.4** | **12.0** | 85.1 | 93.8 | 97.8 | 86.2 | 78.1 |
> | ***GeoMoLa* (Ours)** | **69.4** | 85.9 | **98.4** | **81.4** | 79.0 | **12.0** | **90.5** | **95.0** | **98.2** | **92.1** | **80.1** |
>
>
> **Table R2. Task completion rate on Calvin.**
>
> | Method | Training Data | 1 Task | 2 Tasks | 3 Tasks | 4 Tasks | 5 Tasks | Avg. Length |
> |---|---|---|---|---|---|---|---|
> | **RVT2** | Language-annotated | 90.4 | 76.8 | 61.6 | 50.2 | 39.8 | 3.23 |
> | ***GeoMoLa*** (**Ours**) | Language-annotated | **95.4** | **84.0** | **72.4** | **59.0** | **46.4** | **3.58** |
>
>
> While RVT2 utilizes RGB-D input, it obtains supervision only from ground-truth poses. Our results demonstrate that learning motion patterns enables more accurate action prediction without requiring additional views. The performance drop of RVT2 from multi-view to single-view configuration is consistent with observations in other methods - for instance, **3D Diffuser Actor shows a 56.3% success rate reduction on ''stack blocks'' when transitioning from multi-view to single-view settings [4]**. This occurs because understanding scene geometry and task context from a single static RGB-D image is challenging. Our method addresses this limitation by explicitly learning to predict geometric changes over time, highlighting that capturing dynamic motion in demonstrations is crucial for manipulation tasks, beyond merely relying on static geometry.

---

> ### Author Response · Authors · 2025-11-23
> **Respond to the reviewer (part 2)**
>
> > **Weakness 3** – Terminology and Clarity
>
> We appreciate the reviewer's attention to terminology consistency. To eliminate confusion between "latent action" and "motion latent," we have adopted a more precise naming convention in the revised manuscript. The term "motion latent" now consistently refers to latent codes learned from motion trajectories, while "latent action" has been removed to prevent misinterpretation as embeddings in the robot action space.
>
> This clarification is reflected in the updated Figure 2 and throughout relevant sections, which now more effectively communicate the process of learning and leveraging motion latents.
>
> > **Question** – Motivation for MiniGPT-Based Latent Action
>
> The motivation for deriving latent representations from RGB and language using MiniGPT is to leverage large-scale pre-trained vision-language models for incorporating rich semantic and contextual priors. This enables our model to better interpret the robot's state and language instructions without requiring extensive robot-specific training. Although depth/point cloud data are available, we use MiniGPT to efficiently extract high-level reasoning capabilities, which are then combined with the input depth information to predict future 3D geometry. This design enables **both stronger generalization and 3D-aware supervision** while maintaining a lightweight architecture.
>
> ## **Summary** ##
>
> Once again, we extend our sincere thanks for the reviewer's valuable guidance and constructive feedback. Our manuscript has been significantly strengthened through this process, and all additional results have been incorporated into the revised version. We welcome any further questions and remain available to provide additional information as needed.
>
>
> **Reference**
>
> [1] Ye, Seonghyeon, et al. "Latent action pretraining from videos." arXiv preprint arXiv:2410.11758 (2024).
>
> [2] Bruce, Jake, et al. "Genie: Generative interactive environments." Forty-first International Conference on Machine Learning. 2024.
>
> [3] Wu, Guanjun, et al. "4d gaussian splatting for real-time dynamic scene rendering." Proceedings of the IEEE/CVF conference on computer vision and pattern recognition. 2024.
>
> [4] Ke, Tsung-Wei, Nikolaos Gkanatsios, and Katerina Fragkiadaki. "3d diffuser actor: Policy diffusion with 3d scene representations." arXiv preprint arXiv:2402.10885 (2024).

---

### Official Review · Reviewer_ixNB · 2025-11-10

**Soundness:** 3
**Presentation:** 2
**Contribution:** 2
**Rating:** 6
**Confidence:** 4

**Summary:**

The presented work introduces a novel approach to learn latent actions. Rather than learning the latent actions to predict future images, the authors propose learning the latent action to predict both future RGB and pointclouds. Thanks to the pointcloud prediction, the authors claim that the latent actions better capture the 3D geometric of the task.

Thanks to a better geometric component in the learned latent actions; when exploited for policy learning, the learned policies lead to better policies (higher task success rates).

The performance of the model was evaluated both in simulation (CALVIN, RLBench) and real robot. The authors also present ablation studies on the impact of learning the latent actions with and without pointcloud prediction and with and without RGB prediction.

**Strengths:**

**Originality**
The presented work is original in learning latent action representations with the additional 3D geometric embeddings. While previous works [1] propose learning the latent action embeddings with only future RGB prediction, the presented work proposes learning the embedding with both RGB and pointcloud.
The authors argue that the pointcloud might led to better capturing the geometry, a reasoning that is sound.

**Quality**
The work makes a good job in evaluating the performance of the model under multiple evaluations and present a useful ablation to visualize the real impact of adding 3D geometric prediction in the latent action pretraining.

**Clarity**
The work is easy to read and to follow.

[1] Ye, S., Jang, J., Jeon, B., Joo, S., Yang, J., Peng, B., ... & Seo, M. (2024). Latent action pretraining from videos. arXiv preprint arXiv:2410.11758.

**Weaknesses:**

- Weak improvements due to the 3D geometry. While the authors show in Tab 3., a performance improvement thanks to the 3D geometry, the improvements are not large (2% increase in RL Bench and max 5% in CALVIN). Also the variations on the performance increase among tasks, makes it wonder when does the 3D geometry helps and when does not. Authors could consider exploring some simple “demo tasks”, one where 3D geometry does not help and one where 3D geometry is essential and find out if the latent embeddings with the 3D geometry is useful.
- Another interesting analysis could be done in comparing the performance enhacement individually in tasks that require 3D translations informations and tasks that require rotation information. Is the pointcloud-based latent action embeddings equally useful for both?
- Figure 1 is not very informative. Being the first figure of the paper, authors could try to improve the first figure to give a better grasp of the main idea. While it is able to give the general idea of “3D better”, it lacks details and it is too general to be valuable. Authors could consider including information regarding the latent action embedding and how is different from previous latent action embeddings.
- 3D diffuser actor reported performance is lower than in their paper. While the original paper claim an average success rate of 81.3%, in your work the performance is 77%. Is there a reason for this mismatch?

**Questions:**

- 3D diffuser actor reported performance is lower than in their paper. While the original paper claim an average success rate of 81.3%, in your work the performance is 77%. Is there a reason for this mismatch?

- What could be the reason of observing not very large performance enhancement when training the latent actions with pointcloud prediction?

---

> ### Author Response · Authors · 2025-11-23
>
> We sincerely thank the reviewer for their positive assessment of our work's **originality**, the **quality of evaluation** across multiple benchmarks and ablations, and its overall **clarity**. Their insightful questions have guided us in deepening our analysis.
>
> > **Weaknesses 1 & 2** – Analysis of 3D Geometry Contribution and Task-Specific Benefits
>
> We appreciate the reviewer's suggestion to further analyze the performance gains from 3D geometry. Following this feedback, we conducted additional real-world experiments focusing on cluttered scenes, where geometric understanding is critical. The results are summarized below:
>
> **Table R1. Task success rate on real-world experiments.**
> | Method | Clean cup | Stack cups | Put cups on shelf | Stack cubes | Place dish | Place cube | Average |
> |---|---|---|---|---|---|---|---|
> | GeoMoLa w/o Pointmap| 20.0% | 10.0% | 30.0% | 20.0% |**50.0%** | 80.0% | 35.0% |
> | GeoMoLa (Ours) | **60.0%** | **30.0%** | **40.0%** | **50.0%** | **50.0%** | **90.0%** | **53.3%** |
>
> As shown in the newly added results, removing point map prediction (GeoMoLa w/o pointmap) leads to a significant performance drop— 40% success rate gap—in heavily occluded settings (see ***clean cups***). In contrast, for less cluttered tasks, the gap is smaller (see ***place cube*** and ***place dish***). This confirms that 3D geometry learning is particularly beneficial in complex, occlusion-rich environments. Visualizations of these scenes are provided in Figure 1 (***clean cups***) and Figure 7 in Appendix C.
>
> Additionally, to analyze the role of geometry in different motion types, we picked four representative tasks requiring predominantly rotation vs. translation in RLBench and conducted the ablation study.
>
> **Table R2. Success rates on four tasks on RLBench.**
> | Method | Close jar (trans.) | Push buttons (trans.) | Open drawer (rot.) | Sweep to dustpan (rot.) |
> |---|---|---|---|---|
> | GeoMoLa w/o Pointmap| 69.0 | 91.6| 76.0 | 90.2 |
> | GeoMoLa | **69.4** | **92.1** | **85.9** | **98.4** |
>
>
> In rotation-heavy tasks (e.g., ***sweep to dustpan***, ***open drawer***), removing point map prediction reduced success rates by nearly 9% on average, while in translation-heavy tasks (e.g., ***close jar***, ***push buttons***), the drop was less than 1%. This suggests that 3D geometry is especially important for tasks involving fine rotational control and complex object interactions, aligning with our motivation that geometric awareness supports more precise gripper state estimation and interaction reasoning.
>
> We believe the moderate overall gains (2-5%) occur because not all tasks require detailed 3D geometry. As shown in our new analysis, **the benefit is most pronounced in cluttered scenes or rotation-heavy tasks**. Thus, the key value of our approach lies in providing consistent robustness in geometrically complex settings, rather than a uniform boost across all tasks.
>
> > **Weakness 3** – Figure 1 is not informative
>
> We deeply thank the reviewer for the suggestion and have revised Figure 1 to better illustrate the latent embedding learning process to highlight the differences from previous methods. We also give a clearer view to illustrate the differences in the generated trajectories guided by different latents (driven by 3D or 2D).
>
> > **Weakness 4 & Question** – Performance Differenc e in 3D Diffuser Actor
>
> The performance difference noted by the reviewer stems from the difference in camera settings. The original 3D Diffuser Actor paper reports 81.3% average success under a multi-view setting, whereas our experiments—as stated in Line 309 and consistent with our overall single-view setup—are conducted using only the front-view input. It represents a common and practical configuration for real-world robot systems. All baselines and our method were evaluated under this same single-view configuration to ensure a fair comparison.
>
>
> ## **Summary**
>
> We are deeply grateful for the reviewer's thoughtful comments, which have significantly strengthened our analysis.  We have included the additional results and improved Figure 1 in our revised version. Should any further questions arise, we would be happy to provide additional clarification.

---

> > ### Comment · Reviewer_ixNB · 2025-11-25
> > **Answer to the authors**
> >
> > I thank the authors for the additional experiments showcasing the benefit of adding 3D predictions to the latent action training models. The new experiments are benefitial as previous experiments were not showing major improvements.
> >
> > I observed the Figure 1. I still would encourage the authors to revisit it, as it is very weak in information. By observing it, the only signal of information is "3D better than 2D". I believe the authors could benefit from the figure (being the main figure of the paper) to add additional information, maybe related with latent action models. Consider that this is the figure the reader going to see first. It would be interesting to share the main ideas of the paper there, beyond "3D bettter than 2D".

---

### Meta-Review · Area_Chair_GCRy · 2026-01-07

**Summary:**

The paper explores the use of Geometry-Aware Motion Latents for improving manipulation policies. The paper uses VQ-VAE to learn latent codes.

After going over the reviews and rebuttal, the AC feels that the paper warrants a borderline rating, tending towards rejection. Among the remaining concerns, the AC feels that the empirical justification for the work is limited (as identified in the outstanding concerns). The paper is close to borderline, and the AC encourages the authors to resubmit after addressing the concerns.

**Reviewer Concerns:**

Addressed:
- Stronger RVT2 baseline added.
- 3D Diffuser performance discrepancy.
- Toning down of "First 4D" phrasing.
- Terminology inconsistency and unclear sections.

Outstanding:
- Weak overall improvements from 3D Geometry. The authors provide some arguments, but the evidence feels weak. The AC agrees that the added complexity of a new stage of training is difficult to justify given the limited improvement (<2% on RLBench).
- Another issue the AC identified is why the authors did not include all 20 RLBench tasks as done in PerAct, RVT, or 3D Diffuser Actor.
- Additionally, CALVIN results do not include some more recent methods like RoboUniView. It is fine not to compare with everything empirically, but there should be some justification for these omissions.

**Reviewer Scores:**

Reviewer ixNB -> Likely unchanged
Reviewer ViXr -> Increase from 2 to 4
Reviewer voeQ -> Likely unchanged

---

### Decision · Program_Chairs · 2026-01-26

Reject